# Creating a Healthy Environment for Elderly People in Urban Public Activity Space

**DOI:** 10.3390/ijerph17197301

**Published:** 2020-10-06

**Authors:** Weiting Shan, Chunliang Xiu, Rui Ji

**Affiliations:** Department of Architecture, Northeastern University, Shen Yang 110819, Liaoning, China; xiuchunliang@mail.neu.edu.cn (C.X.); jiruivia@163.com (R.J.)

**Keywords:** urban park, elderly healthy, acoustic environment, planning principle

## Abstract

According to statistics, the global, population aging problem is severe and growing rapidly. The aging problem is most obvious in some European countries, and most of them are developed countries, such as Japan, Italy, Germany, France, etc. The current internal and external environments of parks in China are complex. The inefficient utilization of space in urban parks is a prominent problem. The design of public spaces that only considers the visual experience is incomplete. Based on the optimization of urban park space planning principle, this study examined a new measure of the acoustic environment in elderly public activity space and designed a new elderly healthy urban park environment. Methods: Using the main parks in Shenyang (Zhongshan Park, Nanhu Park, Youth Park, and Labor park) as the study sites, this study analyzed problems in the acoustic environmental data through on-site inspection, questionnaire survey, and physical data collection. By using general linear regression and multiple regression methods, this study analyzed the impacts of plant density, site elevation, structure enclosure, functional mixing degree on the acoustic environment, and elderly population activities. Based on the acoustic environment, we propose improvements and construction ideas, as well as technical methods, for urban elderly public activity space planning. The utility of the “elderly public activity space planning principle” was also considered. Results: Elderly activity space in urban parks was affected by three main factors—plant density, degree of structural enclosure, and function mixing degree. These factors should be optimized to construct healthy acoustic environments and attract different types of people. Discussion: Compared to past studies, the new influencing factors of the planning principle for elderly public activity space found in this study, would benefit the urban park environment for the elderly and support sustainable development of cities. Conclusions: This study proposes three optimizations to the elderly urban park space planning principle and builds four healthy models of elderly urban space activity.

## 1. Introduction

### 1.1. Urbanization Development

Arguments are commonly made that sustainability challenges cannot be addressed effectively using conventional approaches to policy and planning. The development of inhabited centers was the basis of the urbanization process of the city territory, with the creation of new cities and the expansion of existing ones, in a process that continues today and is likely to continue into the future [1,2]. However, these changes are having a serious effect. Social sustainability combines design of the physical realm with design of the social world—infrastructure to support social and cultural life, social amenities, systems for citizen engagement, and space for people and spaces to evolve [3]. Sustainable urban development is closely related to urban ecology. An eco-city secures ecologically sound, socially beneficial, and economically viable development that is supported by planning, design, and transportation [4,5]. It is of great significance for urban ecological protection and sustainable development to study the change in characteristics of blue and green space, during urban expansion [6]. The pressure of urban renewal and limited land area are the main restrictions on the adequacy and quality of public space provided by the block [7,8]; limited urban public space has become a primary problem in urban development. High-density and mixed-use urban planning models usually mean that different space types are closer and more possibilities for outdoor activities occur, enabling people to make better use of public space in the neighborhood and generate more social opportunities [9]. City parks are the green lungs of cities, and also an important way for city residents to maintain contact with nature. In Asia, Japan is the earliest country to go through urbanization. Japan was always at the forefront of the world, in terms of improving the land use rate, and their experience in solving this problem provides a reference. For example, in Nagaoka, the city government became a civic center. In most recent studies, the government functions, citizen interaction, and public services were organically combined into a “citizen-gathered city government”, and the “Aore Nagaoka” commercial complex was revised and built to increase land use rate [10]. Urban green space (UGS) is crucial to the healthy development of urban residents, and UGS that is accessible can benefit residents to an even greater degree [11]. With the development of urbanization in China, urban parks became an irreplaceable urban green space, playing a positive role in the life of residents [12]. Therefore, problems associated with urban green land should be solved to improve the physical and mental health of urban residents.

### 1.2. Ageing Society

Globally, the elderly population group is increasing more than the other age groups, and at a faster rate in developing countries [13]. The elderly is one of the populations that use urban green land more than others like older adults. In 1980, the elderly population represented the majority (56% of persons aged 60 years or over) in developed countries. However, in 2050 almost 80% of the population would be aged 60 years and over in less-developed countries [13]. In 2019, China’s population of age 65 years and over reached 176.03 million, an increase of 9.45 million, accounting for 12.6% of the population, which is 0.7% higher than that in 2018. Japan, Italy, and Portugal are among the top three economies in terms of the proportion of elderly people in the world, and China ranks 61st among global economies [14]. Since 2012, the elderly population in Shenyang (SY) is growing, with the average annual growth rate projected to rise from 4% before 2012 to 7% from 2012 to 2040 [15]. It is necessary to reconstruct elderly urban park space in SY. Developing the best methods to plan elderly outdoor activity space to support their health is becoming an international concern. For example, recent studies from Europe implemented the concept of age-friendly cities in The Netherlands and Poland. The studies illustrated the potential of making cities more appropriate to the needs of older people and identified important challenges for active aging in current and future generations [16]. In high-density cities, such as Macau, many elderly people live in historic urban districts because of the familiar community culture and spatial environment. As older people are less mobile than younger people, they have smaller social and activity radii. In addition, the elderly in Macau spend most of their daily life in the surrounding public space [8]. In recent years, it was acknowledged that urban green space (UGS) is crucial to the healthy development of urban residents. UGS that is accessible can benefit residents to an even greater degree [17]. Urban park space is the main elderly activity space, and elderly people are also the main users of urban parks. 

### 1.3. Noise Problem in China

Dancing, tai chi, and stretching routines are common sights in parks across China, often carried out in groups and accompanied by music from loudspeakers. It should be noted that in the urban park zone, the acoustic environment is greatly affected by the regular activities of tourists, and amplitudes fluctuate over time [18]. There are few studies that included sound as an important environmental factor to study a crowd’s preference for activity space, and it was not used to study the interaction between elderly activity space and the soundscape experience in urban parks. Group dancing, or square dancing, has gained attention at home and abroad, since it is peculiar to China. However, it has also been controversial, with people living near parks complaining about the noise [19]. According to the theory of needs hierarchy proposed by the American psychologist Abraham Maslow in the paper “Human Incentive Theory” in 1943, musical activities satisfy the needs of the elderly for the third and fourth levels (love and belonging) and the requirements of the fifth level (self-actualization) [20,21]. Due to the diverse age structure and physical and psychological structure of various people, some people do not like or even oppose such activities, which generates conflict in the use of urban park space between the elderly and young people. For example, in May 2018, the website Xinmin.com reported that an old man was splashed with water because he was singing in the park, as it led to a conflict with dancing and karaoke television (KTV) outside elderly in space. In November 2019, the Southern Metropolis Daily reported that someone clashed with the dancing elderly over snatching of a venue. It is increasingly prevalent to hear such news, which reflects the conflicts in the use of urban park space. In a study of the adaptability of activity space in the park for the elderly, to improve the overall benefits of the park, it was necessary to comprehensively understand the activities and selection habits of the elderly, so as to make targeted improvements in the spatial layout and detailed design to meet their needs [22]. Urban parks are an important component of UGS, so to plan them without an auditory perspective and only accounting for the aesthetic conception is incomplete [17]. Dance enthusiasts in Guangzhou who like to practice in public parks said that they would obey the city’s pending restrictions on noisy fitness and recreational activities in parks, but they hoped there would not be a total ban on group dancing in public places [19]. Thus, taking the urban acoustic environment as the entry point to create elderly urban park space is necessary in research.

However, existing studies did not place much attention on the acoustic environment in sustainable urban park use. A lack of elderly care and healthcare existed as a health problem for many years, but younger people are believed to hold less stringent views on the environment and have lower levels of environmental concern [20,21]. A positive relationship between physical activity and life satisfaction was found in older adults [23]. Here, we sought to understand the factors that influence elderly urban activity space planning. The urban park plan strategy should clearly define the overall objectives, which should be based on the healthy activity space of the elderly in city parks. Thus, the general goal of this study was to explore the interactions between ‘space’, ‘sound’, and ‘people’. The research questions were “What factors influence elderly urban activity space planning?” and “What factors influence sustainable urban park land use about elderly acoustic environment”. The main issue was to solve space associated with elderly activity based on the acoustic environment. The research model is shown in Figure 1.

## 2. Methodology

### 2.1. Case Study Location

The research area was SY. SY has the largest number of left-behind, lonely, and empty-nest elderly people in China, and it is a post-industrial city located in Northeast China. With the decline of industrialization, the city is confronting many new social problems, e.g., those associated with the ageing population [24]. In the urban park space, the conflicts between elderly and young residents are more prominent than before. Elderly residents are the primary users of SY city parks.

According to a previous investigation, problems associated with the elderly are particularly obvious in the Zhongshan Park, Nanhu Park, Youth Park, and Labor Park [25,26]. Zhongshan Park is located in the south side of Zhonghua Road, Taiyuan Street business district, near the SY railway station and close to the subway line. The proportion of elderly users was 61%. Nanhu Park is located next to the Northeastern University in the Heping District, close to the city’s main road—the Nanhu Road. After a sample survey, it was found that the elderly accounted for 64% of the users of this park. Youth park is located in the central economic corridor of SY city—the golden corridor of the Qingnian Street section, with surrounding office areas, business districts, and large crowds; the elderly accounted for 43% of the users in this park. Labor Park is located in the Zhaogong South Street, Tiexi District; it is surrounded by residential buildings and the number of residents is considerable. The elderly accounted for 62% of its users. The four parks are mainly used by the elderly, and the main activities of the elderly are exercise, such as square dancing and singing, while young adults tend to walk and exercise.

### 2.2. Research Design

In this research we undertook a literature review, field investigation, questionnaire survey, physical survey (sound level meter), general linear regression analysis, and multiple regression analysis. Analysis accounted for the fact that interactions between social and ecological systems are complex [27,28].

In the initial stage of the study, it was necessary to take sound as the entry point, in order to summarize the typical spaces where users conduct music activities in the park, and determine the research area, which helped understand the site selection and construction of sound activity sites. Through a pre-investigation of seven representative parks with dense music activities and prominent conflicts in downtown Shenyang, we included the Zhongshan Park, Labor Park, Nanhu Park, and Youth Park. According to the factors of dynamic or static space, size of space, surrounding structure, and plant density, we classified 8 different characteristic spaces—dynamic and external space, gallery and landscape walls, slightly dense plant; internal space, intersection, few seat, and plant; big size, sink terrain, dense plant; static space, hilly terrain, pavilion, and few plant; big size; static space, external space, feature-rich, dense plant; big size, dynamic and external space, pavilion, seat, and slightly dense plant; small size, dynamic and external space, no pavement, close to fishing ground, and few plants; small size, dynamic and external space, one side with building, and few plants. In the verification stage of the study, it was necessary to study the important factors affecting sound transmission in the region and determine the corresponding influencing factors. The relationship between activity population and different spatial factors was explored in the Nanhu Park, Youth Park, and Labor Park, to gain a full understanding of the physiological and psychological needs of the elderly. The research returned back to the “people” layer in the verification stage, to investigate the phenomena and intensity of human activities, under the influence of the factors identified. This research explored the relationship between the number of people and space. In the initial study of the relationship between impact factors and sound propagation, the four factors were directly analyzed in a regression model using SPSS (Statistical Product and Service Solutions). The parameters included the total number of people that were active, walking, dating, relaxing, using fitness equipment, running, chatting, playing chess, or walking through. We used regression analysis to model the number of people and the proportion carrying out the activity; thus, the relationship among city park, sound, and people was established, and effective spatial planning strategies were proposed.

#### Influencing Factors

Planning principles should be based on the analysis of factors affecting group activities. Specific tasks need to be developed for each discipline, with a focus on how each can build off and integrate with one another [29]. Factors affecting urban green space include nature, economy, and society, among which natural factors are dominant and include geographical location, landform, sunshine conditions, the river and lake system, climatic conditions, hydrological conditions, and natural resources [30]. We studied the urban park space of the elderly from the perspective of natural factors. First, according to Ulrich’s stress reduction theory [31], stress could be reduced by contact with nature. People become tired when they are excited or stressed for a certain period of time, and when they are stimulated by constant excitement, their spirit would be seriously damaged [32]. The natural environment plays a certain role in mitigating this impact, plant density in particular. In this study, the influence factor of “plant density” was used to reflect the richness of plants and the naturalness of the environment. According to Kaplan’s preference matrix theory, the four impact factors included site coherence, complexity, readability, and mystery. The site elevation and degree of structural enclosure could influence site coherence, complexity, readability, and mystery [33]. Thus, in the study of site space, from the perspective of landscape pattern complexity and the enclosure of structures, we included two influencing factors—“site elevation” and “degree of structural enclosure”. According to the theory of adaptation level proposed by Helsen [34], through the comparison of the functional completeness of different activity spaces, the suitable site space type can be found; “functional mixing degree” was included as an influencing factor to reflect this. Based on Bellini’s arousal theory [35], the changes brought about by awakening are related to the environment, and “virescence graceful degree” was used to reflect the quality of the environment, as a research factor of the relationship between space and people. Secondly, “virescence graceful degree” was added together with a total of five influencing factors, to explore the changes in the number of people. Finally, the site space was comprehensively considered and effective planning principles for urban park space were proposed. In addition to service demand and suitability, existing green spaces were given priority [36]. The selection of impact factors was based on the study of the soundscape, according to four impact factors—plant density, site elevation, degree of structural enclosure, and functional mixing degree. Then, virescence graceful degree was added together with a total of four influencing factors, to explore the changes in the number of people. The influence factors are shown in Table 1.

### 2.3. Questionnaire

Questionnaire surveys were conducted for one month in 2018 and two months in 2019. Surveys were carried out 5 days a week, including working days and weekends. Questionnaire 1 was focused on influence factors of plant density, structure enclosure, site elevation, functional mixing degree, and questionnaire 2 was focused on the influence factors of virescence graceful degree and preference. The measurement time was from 7:00 a.m. to 7:00 p.m. The field survey was conducted when the weather was good for outdoor leisure activities of the elderly, so that the music activity site and the characteristics of human activities could be judged [37]. Most of their music activities were long-lasting and frequent, and the purpose of the activities was different. These were mainly based on the pleasure of the body and mind, the display of talent, and interpersonal communication. Most people gave high evaluation to musical activities, as they benefit their physical and mental health and singing and dancing skills; they also met the needs of making friends—65% of participants met more than 10 friends and dance partners during the event.

The questionnaire was conducted on a 5-level scale, and the 5–1 scale was used to represent the five answers of “strongly agree”, “agree”, “average”, “disagree”, and “strongly disagree” to each statement. In view of the above factors, we asked questions including “which aspects of the activity site are you satisfied with? (multiple choice)”, and “what activities do you usually prefer when you come to the park? (multiple choice)”.

### 2.4. Sound Collection

The sound level meter is a conventional instrument for measuring the level of noise [38]. The human ear’s perception of the loudness of a sound is proportional to the logarithm of its intensity. To this end, the sound pressure level is used to represent the size of the sound [38]. For different environments, we selected the measurement location and a reasonable height, while minimizing the impact of the surrounding environment and measurement noise to obtain data [39]. In Shenyang, the high wind during spring has a great impact on the accuracy of measurement results, so this survey did not include measurements from spring. Three days of each month were selected to measure the acoustic environment in the scenic area. In order to make the measured value as close as possible to the sound level that people hear when visiting, the sound level meter was placed at a height of from 1.2 m to 1.5 m during the measurement [37]. In terms of sound measurement, a sound level meter was used at fixed points around the music activity space, with 8 different characteristic spaces, to ensure that the data of sound intensity could be measured, and the layout plan of each activity space could be drawn. The spatial characteristics of 28 musical activities in 3 parks indicated that there were 8 different types of activity space, as depicted in Figure 2. These were used to further analyze the influence of spatial characteristics on sound transmission. According to the acoustic environment map, these 8 different characteristic spaces experience a lot of noise—sound source values reached 90 decibels in the center of all 8 spaces, and the maximum sound source value exceeded 110 decibels at space types A, D, and E. This kind of noise affects other users in the park.

### 2.5. Sound Model

To study the influence of sound intensity on people, the number of participants and passive participants in the same music activity was calculated at different volume levels. To study the influence of sound intensity on crowds gathering in the space, a model was built using statistical methods to intuitively show the attitude of different groups towards the change of sound intensity and formulate planning strategies that can attract people and ensure their comfortable activities within the park space. This was also used to optimize the soundscape [40]. Soundscape mapping includes location, source value, time, pressure level, and attenuation. Based on the questionnaire survey, drawing on relevant theories, the influencing factors were found, and then SPSS regression analysis was conducted, using the measured sound intensity data to establish the model. The general linear regression and multiple regression analysis in SPSS were used to show the correlation and impact of the influencing factors. Four models were obtained using these two analyses methods. The models included a positive sound sensitivity model, negative sound sensitivity model, and sound sensitivity model, which mainly depend on the functional relationship presented by the total fitting line of the SPSS scatter plot, where the constant coefficients were in accordance with the chart. 

#### 2.5.1. Sound Sensitivity Model Construction

The linear regression method was used to establish the model, and 59 samples were extracted. The significance level value of the independent variable was less than 0.001, and the tolerance was 1, which was reasonable. It could be considered that this model could simulate the relationship between people and the positive sensitivity of sound, which is shown in Equation (1): (1)$$P_{e}=C_{2}+n_{1}L_{p}$$
where *P*_e_ stands for number of people, C_2_ is the constant coefficients of people, *L*_p_ is the intensity of sound, *L*_p_∈[60 dB, 100 dB], and *n*_1_ is the constant coefficient not less than 0.

The linear regression method was used to establish the model. It can be considered that this model can simulate the relationship between people and the negative sensitivity of sound; as shown in Equation (2):(2)$$P_{e}=C_{3}+n_{2}L_{p}$$
where *n*_2_ is the constant coefficient of more than 0.

The linear regression method was used to establish the model. The linear relationship between the distribution of the population and the sound pressure level was clear, and was positively correlated with the number of active participants and negatively correlated with the number of passive participants. This indicated that this music form had the opposite effects on the two groups, and clearly reflected that different groups have different needs. This model is shown in Equation (3):(3)$$P_{e}=C_{4}+n_{3}L_{p}$$
where *n*_3_ is the constant coefficient of more than 0.

#### 2.5.2. Sound Intensity Model Construction

We analyzed the relationship between the independent variables and the variable of activity sound. The column of standardized regression coefficients shows the effect of the respective variables on the dependent variable. This is shown in Equation (4):(4)$$L_{p}=\left\{\begin{array}{c} C_{1}+n_{3}H_{e}+n_{4}A_{c}-n_{1}P_{L}-n_{2}W_{a}\;,\;\;\;\;\;\;\;\;A_{c}\in \left[0,\;4\right]\\ C_{1}+n_{3}H_{e}-n_{1}P_{L}-n_{2}W_{a}\;,\;\;\;\;\;\;\;\;\;\;\;\;\;\;\;\;\;\;\;\;\;\;A_{c}\in \left[4,\infty \right] \end{array}\right.$$
where *L*_p_ is sound attenuation; C_1_ is sound constant; *W*_a_ is degree of structural enclosure; *H*_e_ is space elevation; *A*_c_ is functional mixing degree; and *n*_1_, *n*_2_, *n*_3_, *n*_4_ are the constant coefficients not less than 0.

#### 2.5.3. People Number Model

We carried out a multivariate linear regression analysis of independent variables and dependent variables, and the model helped us obtain a graph of partial regression coefficients. The partial regression coefficients of the independent variables *X*_1_, *X*_3_, *X*_4_, and the constant term C showed *P* values <0.05. Under the test level of α = 0.05, it could be considered that the partial regression coefficients were not 0, which was statistically significant; *X*_2_ and *X*_5_ did not meet the above standard, so these two impact factors were discarded from the model. The unstandardized regression coefficients of each term *n*_1_, *n*_3_, *n*_4_, and the corresponding independent variables *X*_1_, *X*_3_, *X*_4_ were multiplied and added to obtain the number model. This is shown in Equation (5): (5)$$Y=C+n_{1}X_{1}-n_{3}X_{3}+n_{4}X_{4}$$
where *Y* is the number of people; C is the constant coefficient of people; *X*_1_ is the plant density; *X*_3_ is the construction enclosure degree; *X*_4_ is the functional mixing degree; and *n*_1_, *n*_2_, *n*_3_, *n*_4_ are the constant coefficients not less than 0.

Since the independent variable *P* (significant level value) greater than 0.05 was not finally included in the regression model, the multiple linear regression equations could be written as
(6)$$Y=6.785+28.192X_{1}-44.895X_{3}+16.101X_{4}$$

## 3. Results

### 3.1. Questionnaire

The four parks studied are located at the center of SY, and are mainly used by the elderly. The main activities of the elderly are exercise, square dancing, and singing. Young adults tend to walk and exercise. According to the questionnaire, the main reasons for participating in music activities were social contact, relaxation, and sports. Questionnaire 1 was mainly distributed to participants in the vicinity and questionnaire 2 was mainly distributed to participants in the vicinity of the music activity. A total of 277 interviewees took this research questionnaire survey, and 250 interviewees contributed valid data (i.e., provided complete answers), accounting for 90.8% of the total number of interviewees. 

According to the statistical analysis of the survey of music activity participants, 79% of the elderly are over 60 years old, and 26% of the participants are elderly people living alone, most of whom have many friends and their own social circle. Among the participants, young people aged 18–40 years accounted for 41% and their voice for volume control was the highest. A total of 45% of the young people showed high or low resistance to loud music activities. People aged 41–60 years accounted for 38% of the sample, while people aged 61–100 years accounted for 21%; the other 2% were minors. The result showed that people aged 41–60 years were the primary park users, while those aged 61–100 years were long-time users. People aged between 40 and 100 years attach great importance to the construction and optimization of urban parks. They hope for there to be an increase of seating facilities and changing rooms, and privacy of music activity space. As shown in Figure 3, the participants mainly take part in quiet activities in the park, and most of them hoped that the park would become quieter.

Space function conflicts between elderly and other residents were particularly evident at Labor Park. The square and pavilion corridor of Zhongshan Park showed high plant density and well-constructed enclosed structures. Most of the venues had converging-space but non-linear space, and the seats and other facilities were complete, so the number of music activities and users was the largest. Youth Park has a beautiful scenery, and the office workers choose to walk, talk, or take a nap in the park during their lunch break, but they are plagued by KTV singing. On the one hand, this conflict reflects that the park does not fully consider the needs of different people in terms of spatial structure and functional planning; on the other hand, it reflects the park’s neglect of sound control in the construction of landscape details. No effective measures were taken to weaken the conflict. We carried out a detailed analysis of the two groups of participants and passive participants to understand the different needs in terms of the soundscape when they perform different activities in the same urban park space. Zhongshan Park belongs to the dynamic balance type; Youth Park belongs to the activity stability type; and Nanhu Park and Labor park belong to the peak value obvious type. Figure 4 shows that people are mostly in a tired state at noon, and they hope not to be disturbed by sound, so the open-air KTV is the most disturbing activity at this time. This is particularly evident at Youth Park and Nanhu Park.

### 3.2. Model Results Analyses

In the acoustic sensitivity model, the aggregation of the two groups under the influence of sound intensity is shown in Figure 4, and the relationship between population distribution and sound pressure level is discussed through mathematical equations. The influence of music type on the two groups of people was the opposite, which clearly reflected that different groups have different functional needs. Notably, questions were framed in such a way that the respondents’ answers reflected how the environment contributes to their soundscape [41].

The investigation was conducted in eight selected typical music spaces to record the number of people around the sound source, and measure the sound pressure level at fixed points. Linear analysis was performed with the number of people (including participants and passive participants) as the dependent variable, and the sound pressure level (sound source’s sound volume) as the independent variable. Analyzing the relationship between the numerical independent variable sound pressure level and the number of people, we obtained a linear relationship between the people and sound from the scatter plot. The results showed that when the sound was louder, fewer people were present who did not engage in music activities; however, the more people who participated in music activities, the louder the sounds were, suggesting that music activities affect sound and indirectly affect park use by other users. This case is depicted in Figure 5.

#### 3.2.1. Sound Intensity Model Results

There is a linear relationship between the degree of structural enclosure and the functional mixing degree, based on the sound intensity model. As shown in Figure 6, plant density, site elevation, and structure enclosure were positively correlated with sound attenuation, while function mixing degree had a negative correlation with sound attention. In other words, people were not sensitive to sound in environments with high plant density, sites with a greater height difference, and closed structures. The influence of sound on non-music activities users was lower in this space. A functional space could be a distraction, reducing the impact of sound.

#### 3.2.2. Number of People Model Results

After linear regression analysis of the number of people model, partial regression coefficients were obtained. As shown in Table 2 plant density, site elevation, and function mixing degree were correlated with the number of people. Combining these findings with sound intensity model results, plant density, site elevation, and function mixing degree were the main influence factors. 

### 3.3. Comprehensive Analysis of Urban Park Space

Through field investigation and the establishment and analysis of the above models, the main problems in urban park space planning were identified. First, the supporting service facilities of activity space were insufficient, resulting in low space utilization rates and unreasonable streams of people, which increased the possibility of conflict. Second, the function in the space was too singular, which led to people ignoring the feelings of others when they carried out music activities, resulting in one group “owning” one piece of land. In addition, the environment of the music space was open, which had a large impact on the surrounding environment. To a certain extent, this also reduced the utilization rate of the surrounding sites, which in turn wasted space. Finally, Figure 7 shows the obvious linear relationships between plant density, site elevation, structural closeness, functional mixing degree, and the number of people. Through regression analysis of the population and five spatial factors, we reached the following conclusions. First, the crowd likes more suitable trees or fewer space activities. Second, because a large number of music activity participants often prefer open squares for gathering activities, due to the small area of the rising or sinking site, the masses often prefer to exercise in lower altitude venues. In addition, people like open sites, and too many enclosures would force people to leave. Finally, in addition to the equipment and collective activities of the music activity participants, people generally preferred spaces with more functions. The results showed that higher the plant density, the more obvious was the site height difference; and the more enclosed the structure, the less people there were. Elderly people did not all like the same space either, as it was not conducive to their communication and entertainment. At the same time, the elderly preferred functional space and the different activities could reduce their sensitivity to sound. Such a space could enable users of all ages to coexist peacefully, reducing land conflicts and conflicts between users in different age groups, as shown in Figure 8.

## 4. Discussion

### 4.1. Planning Principle for Elderly Urban Park Activity Space

Ageing poses many disadvantages to the older adults due to their physical, mental, and cognitive impairment [32]. Over recent years, assistive technologies, such as mobile and wearable sensors, assistive robots, smart homes, and smart fabrics for emergency response were introduced to maintain the independence of older people and to monitor and improve their health condition. Although emergency assistive technologies are useful for older adults, previous research indicated [42] that aged populations, even in a modern country such as Japan, have a more negative attitude towards performing basic life tasks [16]. Therefore, there are still many challenges to planning public activity spaces for the elderly, especially those living alone.

Our results found that the three most important aspects of site design included here were plant density, degree of structural enclosure, and function mixing degree. In terms of the absolute value of the standardized coefficient Beta, degree of structural enclosure was more important than plant density and functional mixing degree. A three-layer design method was adopted to adapt the park design to sound. This study drew out ways to control the transmission of sound. It was proposed that a good sound landscape should be built in the activity space to attract all kinds of people, and the three aspects of plant density, structural closure, and functional mixing should be considered first. Future studies should consider more functions to reflect the full value of such spaces [42].

#### 4.1.1. Plant Density

To a certain extent, plants improved the quality of the environment, which could relieve the excitement fatigue caused by sound stimulation. The attenuating effect of plants on noise was mainly due to their ability to absorb, reflect, and diffract sound waves [43]. Only when the vegetation formed a certain closed green belt community, could the sound energy transmission be effectively blocked [44]. In addition, reasonable urban planning and afforestation could effectively control environmental noise and protect the urban environment [45]. However, our findings suggest that when the plant density exceeded a certain limit, it would form a sort of landscape pressure causing some people to leave, while those who like quiet would stay to meditate, talk, or walk. Based on an analysis of the above results, this study proposed three suggestions for the improvement of urban park space planning, in light of the conflicts that arose among different age groups in different acoustic environments. First, plants should be situated reasonably and spatial function positioning should be clearly defined in the music activity space; we would also recommend the planting of noise-reducing trees. The selection of plants in the gathering area for the elderly should enhance the diversity of colors and ease the fatigue caused by the excitement of the crowd. In addition, trees guarantee a certain degree of canopy closure, which could effectively reduce the adverse effects of excessive sound.

#### 4.1.2. Degree of Structural Enclosure

The traditional way to prevent noise transmission is with a barrier between the sound source and target area to form an audio-visual area behind the barrier, thereby reducing the noise [46]. After statistical analysis, we found that people prefer open spaces and so we are reluctant to build more obstructive spaces for activities. In the process of investigation, only a few people liked to sit quietly in this kind of space for exercise or communication, so this kind of space is more suitable for exercise, playing chess, and other activities that require a “static space”. After research, in terms of building enclosure, plants with certain permeable structures should be used as much as possible in the music area. Through plants of high, medium, and low levels, a landscape wall with sound absorption effect should be formed to realize sound insulation without separating people. 

#### 4.1.3. Functional Mixing Degree

In the process of urban renewal, the function of park green space changes, not only the basic function of green space, but also in carrying out the role of urban cultural inheritance [47]. City parks provide potential functions including sightseeing, viewing, leisure, entertainment, health, fitness, contact with nature, cultural science, and other functions that are conducive to physical health [48]. Another researcher summarized a park’s functions as tourism, ecological service, education, greening and beautification, economic, disaster prevention, hedging, etc. This study showed that the more functional that urban parks were, the more popular they were [49]. At the same time, we should also pay attention to the functional diversity of the same space, equip facilities according to the characteristics of different groups of people, and define different functional spaces.

### 4.2. Planning Principle Optimization

Based on the investigation of the activity habits and the methods of various groups of people, combined with the statistical analysis of the impact of sound intensity, this study proposes a more ideal spatial arrangement for future reference, as shown in Figure 9. The optimized space model is shown in Figure 9 (1). Taking music activities as the spatial center, according to the impact of music activities on different groups of people, the functions from the inside to the outside are chess and card spaces, equipment fitness space, amusement park path, and rest place. Importance should also be attached to the layout of facilities. Static activity venues can ensure that some areas are relatively quiet, so it is necessary to build some landscape walls in such spaces. As shown in Figure 9 (2), the functional space for chat and rest have relatively higher requirements for sound isolation. Thus, the degree of enclosure and area of enclosure should be better than the other sites. As shown in Figure 9 (3), activities like chess and cards, as well as equipment exercise, call for a certain natural environment for shading and blocking certain noise. These spaces have higher requirements for the acoustic environment, with a higher plant density, while the walking space requires a lower plant density. As shown in Figure 9 (4), in terms of the peripheral surrounding function mixing degree, it is important to make full use of all spaces, integrate youth activity facilities in the music activity space, and attract people of different ages to participate in activities. The transitional zone needs an increased sound insulation forest, and can be set up for entertainment facilities according to the terrain, to improve space utilization, while other spaces can be reasonably added according to people’s preferences. Under the premise of reasonable music activities for the elderly, the level of comfort of activities for others should be guaranteed.

## 5. Conclusions

We researched the acoustic environment problems in parks associated with elderly people’s activities in city parks and our research findings would help change the urban park space planning principles. To maintain the health and happiness of elderly people living in the city, their activities in parks should be protected and we proposed ways to solve the problems associated with the noise these activities make in parks. Although the data used in this study were based on large-scale surveys and statistical analysis, and the research conclusions were statistically significant, the parks in this case were all located in a northern city of China. Whether the research conclusions could be used for reference in other parks in southern cities needs further research. In addition, the layout of the park’s site should not only consider influencing factors such as sound, space, and the number of people; other factors should be taken into consideration. Finally, parks and other urban public spaces have different functional characteristics, so the results are not necessarily generalizable to other locations in cities.

In the follow-up to this research, different types of city parks need to be added, and an in-depth study about urban park space planning principles is required. The future research objectives are to separate urban park planning, and creating peace-seeking and quiet urban park space for working people. At the same time, a future study would focus on the different psychological needs of different age groups. The future research direction of urban park space planning principles should be for young people, to help residents of all ages live happily and healthily in the city, and maximize the value of urban park land.

## Figures and Tables

**Figure 1 ijerph-17-07301-f001:**
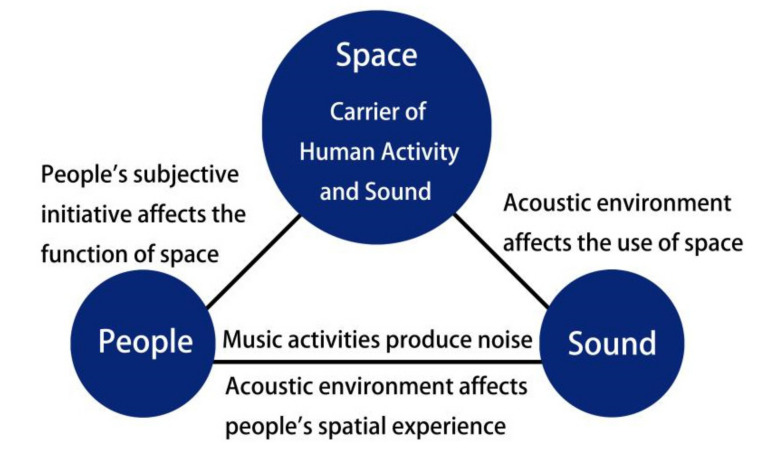
Relationship model of people, sound, and space.

**Figure 2 ijerph-17-07301-f002:**
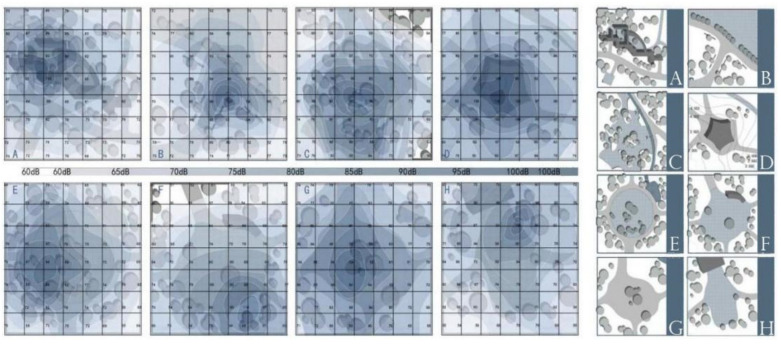
Eight characteristic spatial features and their acoustic environment. (**A**) Dynamic and external space, gallery and landscape walls, and slightly dense plants. (**B**) Internal space, intersection, few seat and plant; (**C**) Big size, sink terrain, and dense plants. (**D**) Static space, hilly terrain, pavilion, few plant; (**E**) Big size, Static space, external space, feature-rich, and dense plants. (**F**) Big size, dynamic and external space, pavilion, seat, and slightly dense plants. (**G**) Small size, dynamic and external space, no pavement, close to fishing ground, and few plants. (**H**) Small size, dynamic and external space, one side with building, and few plants.

**Figure 3 ijerph-17-07301-f003:**
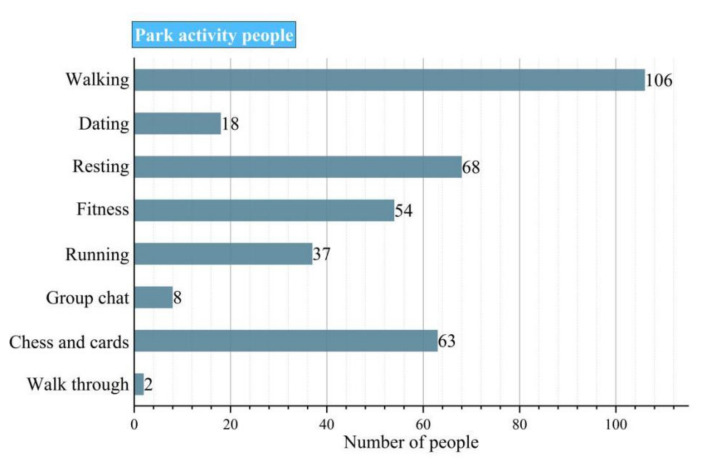
Activity preference (without sound activity).

**Figure 4 ijerph-17-07301-f004:**
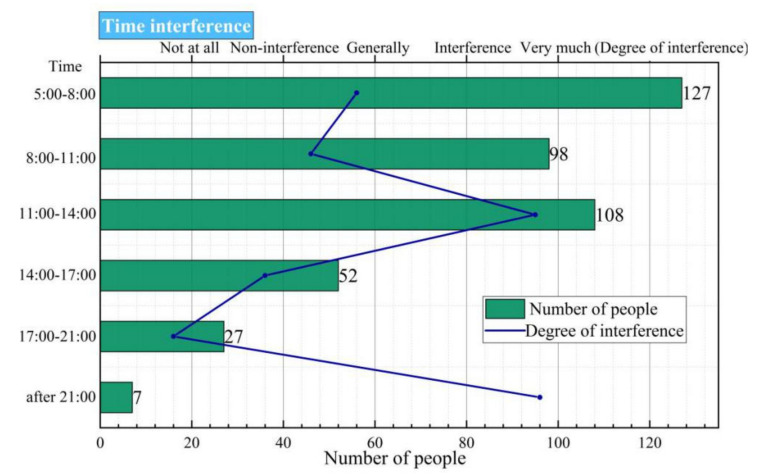
Activity time (without sound activity).

**Figure 5 ijerph-17-07301-f005:**
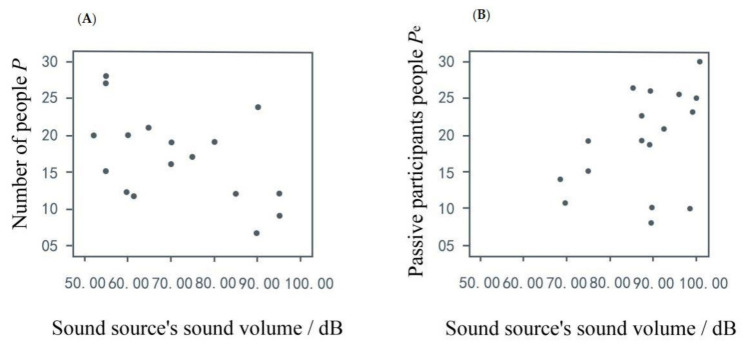
People and sound volume scatter diagram. (**A**) Sound volume has a negative relationship with the number of people. (**B**) Sound volume has a positive relationship with the number of participants.

**Figure 6 ijerph-17-07301-f006:**
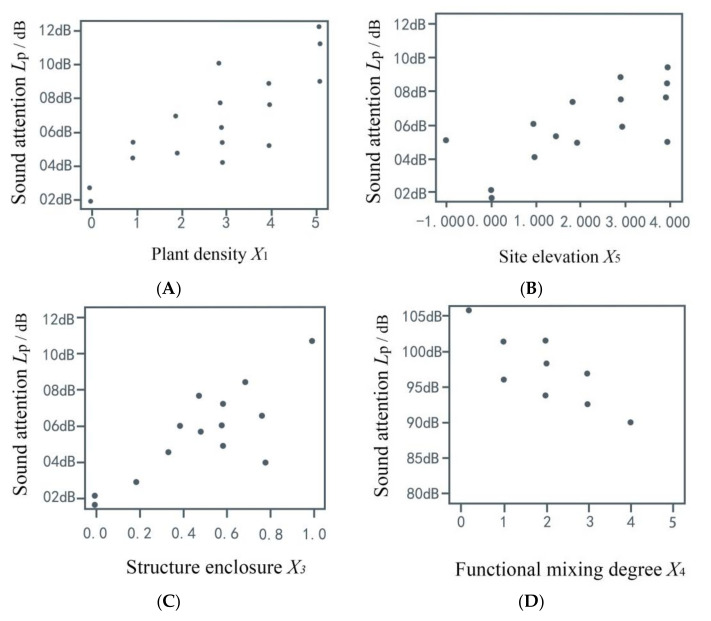
Sound intensity model diagram. (**A**) Plant density has a positive correlation with sound attenuation. (**B**) Site elevation has a positive correlation with sound attenuation. (**C**) Degree of structural enclosure has a positive correlation with sound attenuation. (**D**) Functional mixing degree has a negative correlation with sound attenuation.

**Figure 7 ijerph-17-07301-f007:**
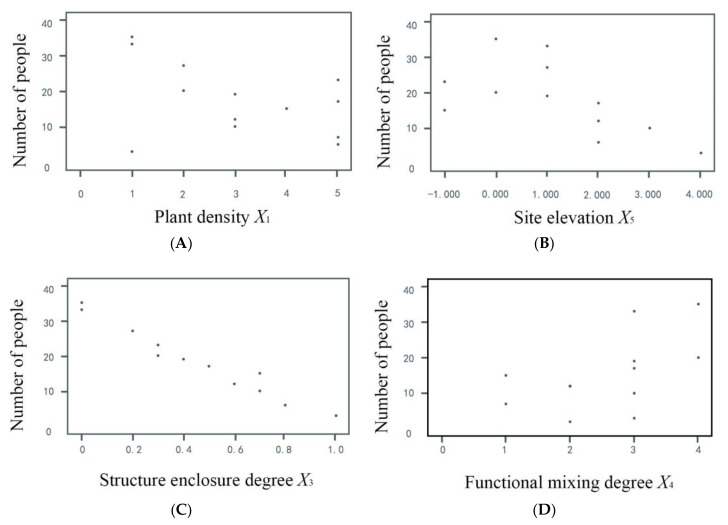
Site conditions and number of people. (**A**) Plant density is positively correlated with number of people. (**B**) Site elevation is positively correlated with number of people. (**C**) Degree of structural enclosure is positively correlated with number of people. (**D**) Functional mixing degree is negatively correlated with number of people.

**Figure 8 ijerph-17-07301-f008:**
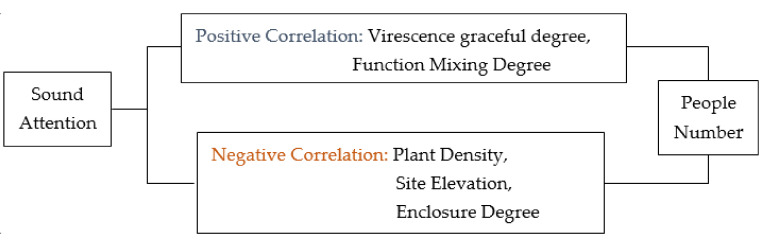
Framework of the “number of people model”.

**Figure 9 ijerph-17-07301-f009:**
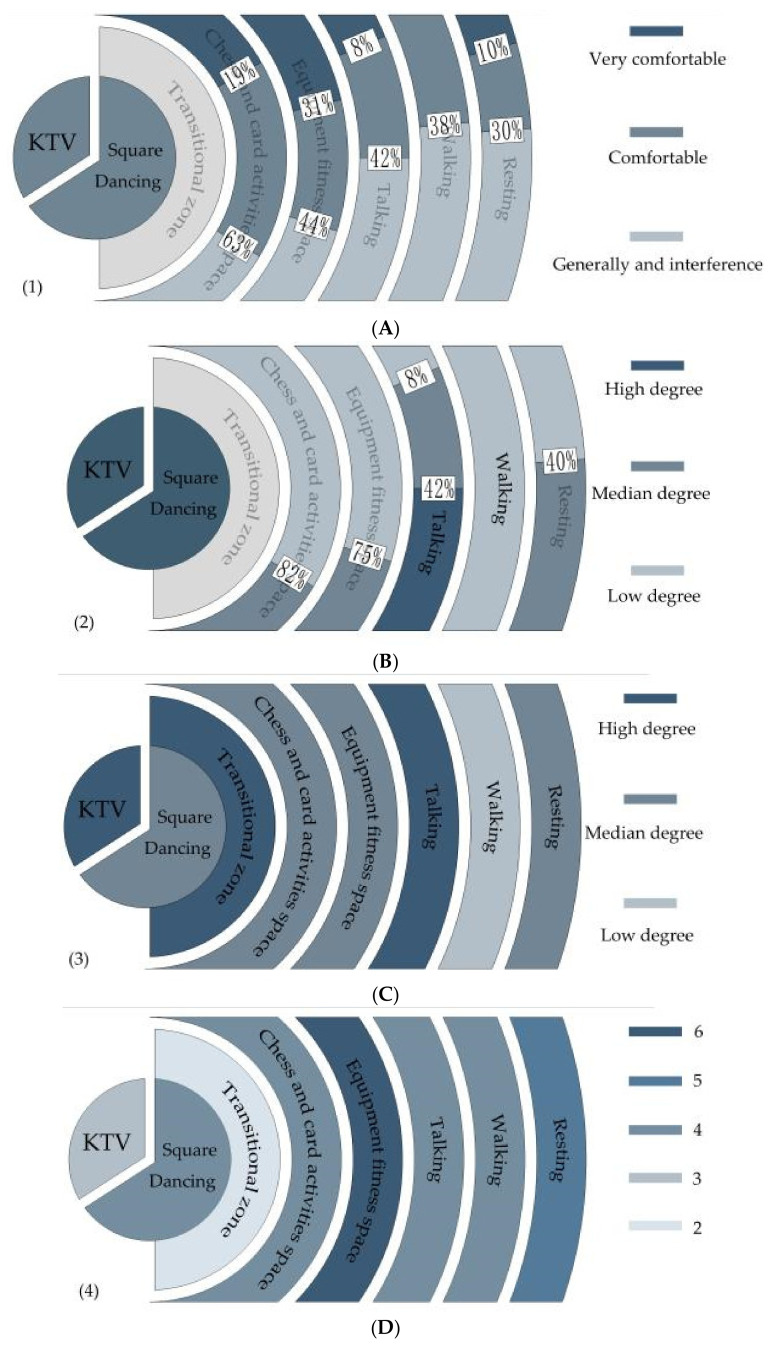
Idealized spatial layout model for a park. (**A**) Space layout, (**B**) degree of enclosure layout (**C**), plant density layout, and (**D**) functional degree layout.

**Table 1 ijerph-17-07301-t001:** Influencing factors used in this study.

Author	Theory	Factor
Ulrich	Stress Reduction Theory	Plant Density
Kaplan	Preference Matrix Theory	Structure Enclosure, Site Elevation
Helson	Adaptation Level Proposed	Functional Mixing Degree
Bellini	Arousal Theory	Virescence Graceful Degree

**Table 2 ijerph-17-07301-t002:** Partial regression coefficient table.

Coefficient ^a^							
Model	Unstandardized Coefficient		Standardized Coefficient	t	Significance	Collinear Statistics	Collinear Statistics VIF
	B	Standard Error	Beta			Tolerance	
C	6.875	8.800		0.781	0.464		
*X* _1_	28.192	10.662	0.628	2.644	0.038	0.031	9.084
*X* _2_	10.757	6.243	0.193	1.723	0.136	0.139	7.716
*X* _3_	−44.895	6.323	−1.209	−7.100	0.000	0.061	8.514
*X* _4_	16.101	5.429	0.320	2.966	0.025	0.151	6.644
*X* _5_	0.850	0.673	0.097	1.262	0.254	0.300	3.335

^a^ Dependent variable: *X*_1_: plant density, *X*_2_: virescence graceful degree, *X*_3_: degree of structural enclosure, *X*_4_: degree of functional mixing, *X*_5_: site elevation/m, and C: constant.

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
