# Peer review of "Creating a Healthy Environment for Elderly People in Urban Public Activity Space"

_ijerph, 2020, doi:10.3390/ijerph17197301_

Round 1

Reviewer 1 Report

This looks like a unique, interesting and well-timed paper, which deals with the best way to plan outdoor activities for older adults based on the acoustic environment. As due to Covid-19 many activities are moved to the outdoor public spaces, making these more welcoming to older adults is very important. However, there is still work to be done, especially in the way the paper is constructed and written.

Abstract

  1. Could the authors start with a little background and context so the reader will understand the importance of this research?

Introduction

  1. Many paragraphs are packed with more than one topic, which makes the reading harder (especially the 1st and 2nd paragraphs). I suggest the authors do re-edit the introduction so reading will be smoother. The reader should understand right from the beginning what is the main issue here. The authors raise a lot of issues, and I suggest to focus only on the main ones.
  2. I believe that as older adults are the primary concern of this paper, they should be mentioned earlier in the paper
  3. The authors opened the 4th paragraph with the sentence ‘’ In the next section, a short review of sustainable land use is given with special regard to case study’’. I advise the authors to connect better the 3rd and the 4th paragraph, and refrain from explaining to the reader what will be included in the paragraph they will shortly read. In addition, the 4th  paragraph should include more MILOT KISHUR
  4. Line 64 - I can understand why the authors gave example from Asian countries, but why from Ghana? And if so, could they give more examples from other countries as well and compare them to the situation in Shenyang? Again, the authors move from example to example without using conjunctions, which makes the reading harder.
  5. A clear section which explains about the situation in China is needed.

Methodology

While it is clear the authors invested a lot of time and effort to choose and implement the best research method for this study, this section requires a rigorous edit.

  1. Line 113 – The authors explain about Kaplan’s Stress Reduction Theory, and give an example of music activity. Right afterwards, the authors write about the ‘’plant density’’, and there is not a clear connection.
  2. Line 119 – the authors further rely on Kaplan’s Stress Reduction Theory and write about its three factors, but later on focus merely on the ‘’complexity’’ factor. Could they explain why?
  3. Line 132 – I suggest to the authors to add a table with all of the influencing factors. This will make it easier for the reader to grasp all of them.
  4. Could the authors explain more in-depth why Shenyang was chosen as a case study? Were there other cities who were considered as part of this study? How Shenyang is similar and different than other areas (besides the % of older adults visiting it’s parks). This is especially important because of the proximity of the parks to the Authors’ University, in addition to possible generalisability issues both in China and worldwide.
  5. Line 134 - How the authors decided which parks to investigate in the initial stage and which in the verification stage to prevent any biases?
  6. Line 141 – the authors are advised to differentiate between all of the research tools. I agree that a short summary of all the tools is needed, but the authors also elaborate more (e.g., about the qualitative questionnaires) where it should be written in the next section. In addition, could the authors cite other works who used the same tools?
  7. The authors wrote that a sound level meter was used to measure at fixed points aroundeight music activity venues (line 159). Why eight?
  8. Line 167 – the authors’ recommendation of using more functions is part of the discussion and not the results. In addition, the next lines are part of the discussion /conclusion as well.

Results

Again, all explanations and justifications of the methods and the research tools should be part of the method. In this section, the authors are advised to focus solely on the results. Data such as survey time and effective rate are part of the method as well.

  1. I wonder if the authors can compare between the 18-40 years old and the older adults - e.g., in figure 3 it will be interesting to see the different activities according to age group
  2. Line 307 – authors should state again the city where the parks are located.

Discussion

The discussion lack reference to other works, and focus merely on analysing the results of the study. Are there any similar works (even from other countries) who tried to explore some of the issues raised in this paper? In addition, The authors should avoid as much as possible mentioning statistics here (e.g, ‘’p<.05, standardised beta). Instead of these, the authors are encouraged to focus on the conclusions. It’s fine to repeat some of the results, but only in words (e.g ‘’The structure enclosure degree is greater than the plant density and functional mixing degree’’). The authors are also encouraged to focus on the effect size of the results, rather than the significance of the statistical analysis.

To sum, this paper is unique, fits the journal’s aims and with interest to the readership, but more work needs to before it could be considered suitable for publication.

Author Response

Dear Reviewer:

Journal: International Journal of Environmental Research and Public Health (MDPI)

Section: Environmental Science and Engineering

Manuscript ID: ijerph-930285

Title: Planning Strategy of Elderly Urban Public Activities Based on Acoustic Environment

Thank you for your review of 10-09-2020. Thank you for providing me with the opportunity to modify and detailed suggestions for modification, which provided me with valuable directions for modification. I am very grateful to your comments for the manuscript. As your extensive editing of English language and style required, we asked for language editing department by MDPI to revise the paper at 19-09-2020, before it was submitted to the journal. I have changed the title to “Creating a Health Environment for Elderly People in Urban Public Activity Space”. According with your advice, we amended the relevant part in manuscript. Here are my responses to your comments. Every of your questions were answered below.

I have revised the manuscript accordingly and the revised portion is marked in red. I hope this will make it more acceptable for publication. We hope that the revised manuscript has addressed all the criticisms raised by the reviewers and that the manuscript is now suitable for publication in Journal of International Journal of Environmental Research and Public Health.

Yours sincerely,

Weiting Shan

Reviewer 2 Report

Dear Authors,

Your paper deals with an important topic of urban planning in growing (mega)cities: the planning of tailor-made and public green spaces referring to different target (age) groups free of user conflicts.

I thoroughly read your paper. In the following, please find my comments and recommendations.

Title of paper: The title of the paper is misleading.

Abstract: The abstract is unstructured. According to the journals requirements please structure the abstract as follows: background of study/methods/results/discussion/conclusion. Please, insert information on the case study parks.

Keywords: Some of the keywords repeat terms already being used in the title of the paper.

Introduction: Due to the complexity of the topic being dealt with, it is necessary to restructure the introduction and therefore, to devide it into subsections such as “growing (mega)cities and urban green space planning”, “functions of urban parks, user conflicts and design planning”, “soundscape concepts for urban parks”. Please insert more references (i.e. related to the soundscape concept of China (line 84).

Line 64 to line 69: I fail to see the necessity to refer to land use planning in Ghana.

In my opinion, it does not become clear that your paper deals with conflicts between the requirements of two different park user groups: elderly park users and young(er) park users.

Line 94: Here, the authors present the “general goal” of the paper. The research questions are missing.

Line 98: Please, redesign Figure 1, bringing together the illustration and the content of the accompanying text box.

Methodology: This section is confusing, mainly due to its lack of structure and the mixing-up methodology and results. (This also applies for the results section which contains information on the applied methodology.). A division into subsections (i.e. research design and research context (research project/third-party funding or rather self-research?), questionnaire, selection of participants, sociodemographic profile of participants (age!)

Line 101ff: Case study location: Please, describe the location context: city (name, number of inhabitants, growth rate, population density, number of parks, …), number and reasons for choice of (4?) parks, selection criteria. This paragraph contains results, but lacks references. Please, insert references.

Line 110ff: Data Collection and Procession: This section is confusing. You present different theories which relate to the topic of your paper and which seem to be relevant for modelling and the discussion of the empirical findings. Nevertheless, you do not cite relevant literature relating to the theories (i.e. “Bellini’s arousal theory” (line 126). I think, a table breaking down the scopes, criteria, measurement and the intersections of the theories would increase readability. Moreover, explanations of specific terms (i.e. relating to Kaplan’s Preference Matrix Theory, cf. line 120f) are necessary.

Relevant details on data collection (year, season, …) are missing (cf. line 197f).

Line 134 to line 140: Please, shift this paragraph at the beginning of the subchapter.

Please, provide a graph of the research design or rather the procedural/methodological steps (cf. line 141ff).

Line 143f: This statement is not (sufficiently) integrated in the content of this paragraph.

Line 146 to line 155: Please, present the questionnaire in an appropriate manner.

Line 156: What does the abbreviation “KTV” mean? Please, explain.

Line 160f: This statement is not (sufficiently) integrated in the content of this paragraph.

Results: Much of the content presented in this section need to be shifted to the methodology section.

The results are presented in a quite “charmless” manner. Since data on 227 park users are available, an appropriate presentation of the quantitative results (frequencies, …) is necessary.

Line 200: What do you mean by “valid questionnaires”, what by “effective rate”? Please, explain.

Moreover, only in subchapter 3.1. it becomes clear, that you surveyed young(er) park users. (Am I right?)

If you undertook surveys in four selected parks, a comparing presentation of findings is missing.

Line 212f: Figure 2: Confusing and is not integrated in the text.

Figures 3 and 4: Please, rename the captions.

Line 321ff: Model Building = Methodology

Please, insert the sources of the referred/selected models and explain to the readers the reasons for choosing the certain models.

Line 305ff: In my opinion, here the result section starts.

Line 317: What do you mean by “the crowd”? Please, explain.

Discussion: This sections lacks referring literature. Please, add.

Line 331: In my opinion, this subsection deals with “planning principles”, rather than “planning strategies”.

Line 393: This section deals with conclusions. Please, rename.

The conclusion section needs content expansion such as the (future) provision of separate/alternative urban parks/green spaces for young(er) busy people seeking peace and quiet.

References: Please, add further relevant literature.

All the best!

Author Response

Dear Reviewer:

Journal: International Journal of Environmental Research and Public Health (MDPI)

Section: Environmental Science and Engineering

Manuscript ID: ijerph-930285

Title: Planning Strategy of Elderly Urban Public Activities Based on Acoustic Environment

Thank you for your review of 12-09-2020. Thank you for providing me with the opportunity to modify and detailed suggestions for modification, which provided me with valuable directions for modification. I am very grateful to your comments for the manuscript. As your extensive editing of English language and style required, we ask for language editing department by MDPI to revise the paper at 19-09-2020, before it was submitted to the magazine. According with your advice, we amended the relevant part in manuscript. Here are my responses to your comments. Every of your questions were answered below.

I have revised the manuscript accordingly and the revised portion is marked in red. I hope this will make it more acceptable for publication. We hope that the revised manuscript has addressed all the criticisms raised by the reviewers and that the manuscript is now suitable for publication in Journal of International Journal of Environmental Research and Public Health.

Yours sincerely,
Weiting Shan

Round 2

Reviewer 1 Report

I would like to thank the authors for their rigorous and quick edit of the manuscript. The current version reads more fluently, and most of the issue I raised were addressed promptly. There are a few minor issues I suggest the authors to address before this manuscript can be accepted for publication:

Abstract

  1. Could the authors start the abstract with a more global statement so the reader will feel this paper might relate to his country as well?

Introduction

  1. Better connection between the different parts is needed. For example, the 1.2 ageing society section could be started with a statement such as: ‘’one of the populations which use urban green land more than others are older adults’’’.
  2. Line 121 – ‘’provide much attention about elderly acoustic environment in sustainable urban park land use’’ is not a research question. Can the authors rephrase?
  3. Line 135 – ‘’ According to a previous investigation, problems associated with the elderly are particularly obvious in Zhongshan Park, Nanhu Park, Youth Park and Labor Park’’ – could the authors give the reference for this investigation?

Design

  1. Figure 2 – Thank you for adding this figure. I suggest removing the type of analysis
  2. Line 221 – the authors write about questionnaire 1 and 2 but the reader does not know what the difference between them and what they included.
  3. Lines 219 and 225(and forward) – please move all the results to the ‘’results’’ section.

Results

  1. Line 321 – ‘’ The retirement age in China is 60’’ – this sentence looks out of context.

Thanks again for the authors for responding to my concerns. 

Author Response

Dear Reviewer:

Journal: International Journal of Environmental Research and Public Health (MDPI)

Section: Environmental Science and Engineering

Manuscript ID: ijerph-930285

Title: Creating a Healthy Environment for Elderly People in Urban Public Activity Space

Thank you for your review of 29-09-2020. Thank you for providing me with the opportunity to modify and detailed suggestions for modification, which provided me with valuable directions for modification. I am very grateful to your comments for the manuscript. According with your advice, we amended the relevant part in manuscript. Here are my responses to your comments. Every of your questions were answered below.

I have revised the manuscript accordingly and the revised portion used the "Track Changes" function in Microsoft Word. I hope this will make it more acceptable for publication. We hope that the revised manuscript has addressed all the criticisms raised by the reviewers and that the manuscript is now suitable for publication in Journal of International Journal of Environmental Research and Public Health.

Yours sincerely,

Weiting Shan

Reviewer 2 Report

Dear Author,

You thoroughly revised the paper, which in my opinion is now ready for submission.

All the best!

Author Response

Dear Reviewer:

Journal: International Journal of Environmental Research and Public Health (MDPI)

Section: Environmental Science and Engineering

Manuscript ID: ijerph-930285

Title: Creating a Healthy Environment for Elderly People in Urban Public Activity Space

Thank you for your review of 29-09-2020. Thank you for providing me with the opportunity to modify and detailed suggestions for modification, which provided me with valuable directions for modification. I am very grateful to your comments for the manuscript. According with your advice, we amended the relevant part in manuscript. Here are my responses to your comments. Every of your questions were answered below.

I have revised the manuscript accordingly and the revised portion used the "Track Changes" function in Microsoft Word. I hope this will make it more acceptable for publication. We hope that the revised manuscript has addressed all the criticisms raised by the reviewers and that the manuscript is now suitable for publication in Journal of International Journal of Environmental Research and Public Health.

Yours sincerely,

Weiting Shan

This manuscript is a resubmission of an earlier submission. The following is a list of the peer review reports and author responses from that submission.